# Exploration of User’s Mental State Changes during Performing Brain–Computer Interface

**DOI:** 10.3390/s20113169

**Published:** 2020-06-03

**Authors:** Li-Wei Ko, Rupesh Kumar Chikara, Yi-Chieh Lee, Wen-Chieh Lin

**Affiliations:** 1Department of Biological Science and Technology, College of Biological Science and Technology, National Chiao Tung University, Hsinchu 300, Taiwan; rupesh.bt01g@g2.nctu.edu.tw; 2Center for Intelligent Drug Systems and Smart Bio-Devices (IDS2B), National Chiao Tung University, Hsinchu 300, Taiwan; 3Institute of Bioinformatics and Systems Biology, National Chiao Tung University, Hsinchu 300, Taiwan; 4Drug Development and Value Creation Research Center, Kaohsiung Medical University, Kaohsiung 807, Taiwan; 5Department of Computer Science, National Chiao Tung University, Hsinchu 300, Taiwan; be341341@gmail.com

**Keywords:** electroencephalography (EEG), brain–computer interface (BCI), working memory, steady-state visual evoked potential (SSVEP), mental focus state, lost-in-thought state

## Abstract

Substantial developments have been established in the past few years for enhancing the performance of brain–computer interface (BCI) based on steady-state visual evoked potential (SSVEP). The past SSVEP-BCI studies utilized different target frequencies with flashing stimuli in many different applications. However, it is not easy to recognize user’s mental state changes when performing the SSVEP-BCI task. What we could observe was the increasing EEG power of the target frequency from the user’s visual area. BCI user’s cognitive state changes, especially in mental focus state or lost-in-thought state, will affect the BCI performance in sustained usage of SSVEP. Therefore, how to differentiate BCI users’ physiological state through exploring their neural activities changes while performing SSVEP is a key technology for enhancing the BCI performance. In this study, we designed a new BCI experiment which combined working memory task into the flashing targets of SSVEP task using 12 Hz or 30 Hz frequencies. Through exploring the EEG activity changes corresponding to the working memory and SSVEP task performance, we can recognize if the user’s cognitive state is in mental focus or lost-in-thought. Experiment results show that the delta (1–4 Hz), theta (4–7 Hz), and beta (13–30 Hz) EEG activities increased more in mental focus than in lost-in-thought state at the frontal lobe. In addition, the powers of the delta (1–4 Hz), alpha (8–12 Hz), and beta (13–30 Hz) bands increased more in mental focus in comparison with the lost-in-thought state at the occipital lobe. In addition, the average classification performance across subjects for the KNN and the Bayesian network classifiers were observed as 77% to 80%. These results show how mental state changes affect the performance of BCI users. In this work, we developed a new scenario to recognize the user’s cognitive state during performing BCI tasks. These findings can be used as the novel neural markers in future BCI developments.

## 1. Introduction

The brain–computer interface (BCI) is a promising communication pathway that translates brain signals into a command to control a device [1]. In past years, the BCI technology has gained increasing attention and has great potential to improve the quality of patient’s life suffering from severe motor disabilities, such as cerebral palsy and paralysis [2]. The BCI system has attracted more attention in artificial intelligence, reinforcement learning, and clinical applications, such as stroke, muscular dystrophies, and spinal cord injuries [1,2]. Despite the significant development of BCI system, there are still some problems in current BCI technologies, like how to recognize that the cognitive state of BCI user is mental focus or lost-in-thought state. These cognitive states changes affect the performance of BCI technologies, like robotic arm for stroke patients, wheelchair and drone. Therefore, the identification of mental focus state and lost-in-thought state is an important issue in the BCI technologies. The steady state visually evoked potentials (SSVEP) are EEG signals has been acquired from continuous visual stimulation at a specific frequency in occipital lobe of the brain [3,4]. In this work, we used two type of SSVEP flicker frequencies at 12 Hz and 30 Hz to understand the effect of low SSVEP flicker stimulation and high SSVEP flicker stimulation in occipital lobe of the brain. The SSVEP elicited by 12 Hz or 30 Hz flashing frequencies, which revealed neural activities changes over occipital lobe of the brain under attention state (mental focus) [5].

The steady-state response to changes in the EEG caused by stimuli, such as the visual stimulus applied to the subject through a computer screen is called the SSVEP. This feature of the EEG signal can be used to assistive in BCI devices for patients to improve their quality of life. A promising approach to BCI technologies uses SSVEP to extract control information. The main advantages of the SSVEP-BCI are a simple and low cost configuration, little effort to adjust the system parameters to the users and relatively high information transfer rates (ITR). Therefore, to investigate the brain dynamics of BCI users in mental focus and lost-in-thought states, we designed a novel experimental scenario combined with SSVEP-BCI, working memory, and distraction tasks. In mental focus state, a subject was fully focused on SSVEP flashes and working memory images when performing a BCI task. In lost-in-thought state, a subject was focused on SSVEP flashing stimuli and working memory images, but they were thinking about something else and forget to respond to the images of working memory due to mind-wandering state. The mental focus and lost-in-thought states are presented in our daily life activities, such as education, research, business, driving, and machine operation [6]. The past studies have been reported that the ability to maintain attention is critical with sustain behavioral goal tasks, like when using the computer or reading a book, people were distracted to think about social websites: Facebook, WhatsApp, Instagram, Twitter, and games. For that reason, people often experienced deviations in attention or attention drift when reading a book, using a computer, and operating a machine [7,8]. Usually, people experienced focus loss episodes, when they performed a BCI task and suddenly realized that they were thinking about something else (i.e., lost-in-thought or mind wandering) [7,8]. People realized that they were in the mind-wandering state and suddenly turned their attention back to work. Mind wandering episode has been related to the appearance of task-unrelated thoughts, and it affects the human focus (i.e., attention) away from the task [7,8]. In addition, lost-in-thought state has been related to low level of alertness and vigilance [9,10]. The high level of attention and willingness to complete a BCI task prevent subjects from falling into a lost-in-thought or mind-wandering state. Previous study reported that the performance of a subject in the working memory task or BCI task depends on the level of attentiveness and level of mental focus [11]. The n-back working memory task has been widely used to investigate the neural mechanism of visual attention [12]. In this study, first time we combined n-back working memory task with SSVEP-BCI and distraction tasks, to investigate the neural activities of mental focus and lost-in-thought states. In the current working memory task, the correct answer of each participant was considered as the ground truth to verify if a participant was in mental focus state or lost-in-thought state. The previous EEG study reported that theta (4–7 Hz) band power increased in the frontal lobe of the brain when participants responded correctly in the working memory task. This finding shows that all participants were in mental focus state [6]. 

Moreover, in previous studies, the SSVEP power has been increased in the occipital and parietal lobes during attention state [13,14,15,16]. The SSVEP power variation depends on the frequency of the visual stimulation and mental focus of the subjects, because some frequency-tagged cortical networks are often sensitive to visual stimulation and others are relatively insensitive to visual stimulation. For example, in one previous study with stimuli reversing at 2.4 Hz and 3 Hz, the harmonic response at 4.8 Hz and 6 Hz was modulated by visual stimulation, whereas the response at 9.6 Hz and 12 Hz was not modulated by visual stimulation [17]. In addition, another study investigated the suppression of SSVEP power using 12 Hz flashing frequency in the prefrontal lobe of the brain in the attention task [18]. However, some previous studies reported an increase in SSVEP power at occipital lobe of the brain with 12 Hz and 30 Hz SSVEP flashing frequencies, which was associated with attention state [19,20,21].

The frontal lobe of the brain has played a functional role in sustained attention [22]. Furthermore, a preceding study reported that the frontal lobe has been linked to the maintenance and development of working memory [23]. Previous studies of EEG signals described that decreased beta (13–30 Hz) band power has been linked to visual attention (mental focus state) [24,25,26]. The SSVEP-BCI study investigated that the occipital lobe of the brain shows a significant role in SSVEP-BCI technologies [27]. Early studies of the working memory task found that the EEG power of the beta (13–30 Hz) band increased in the parietal and occipital lobes [23,28]. Additionally, decreased EEG activity of beta (13–30 Hz) band power in the occipital lobe has been linked to attention (mental focus state) [24,25]. Furthermore, previous EEG studies investigated that increased delta (1–4 Hz) band power in the occipital lobe of the brain has been associated with decreased attention level [29,30,31,32,33,34]. 

The EEG is a non-invasive method, which was widely used in SSVEP-based BCI technologies [1,2,35,36]. Some previous BCI studies reported that the EEG method is capable for recognizing spatial resolution at the human brain [37,38,39]. In this study, we used high temporal resolution electroencephalography (EEG) method to investigate the mental focus and lost-in-thought states of the BCI users. The first time our study explored the EEG neural marker including delta (1–4 Hz), theta (4–7 Hz), alpha (8–12 Hz), and beta (13–30 Hz) bands in mental focus and lost-in-thought states at frontal and occipital lobes of the brain than in previous studies of SSVEP and EEG [6,37,38,39]. According to our hypothesis, the power spectrum of EEG signals in mental focus state will increase more than in lost-in-thought state at the frontal lobe, because the frontal lobe is related to attention and working memory. The SSVEPs power in mental focus state will increase more than in lost-in-thought state at the occipital lobe of the brain, as the occipital lobe is associated with visual stimulation. Generally, SSVEP-BCI study reported that high-frequency stimulation is more comfortable for the BCI user. Accordingly, we hypothesize that the 30 Hz flashing frequency will be more comfortable for the subjects compared to the 12 Hz flashing frequency [40]. Our study investigates the EEG neural markers of mental focus state and lost-in-thought state. These neural markers can be applied to recognize the mental focus and lost-in-thought states of BCI users.

## 2. Materials and Methods

### 2.1. Participants

Thirteen adult volunteers (five females and eight males) aged from 18 to 25 (mean age 22.6 years) participated in this experiment. None of them were taking medication or had a history of psychiatric disorders or brain diseases. All had normal vision. This study was carried out in accordance with the recommendations of the Institutional Review Board (IRB) of National Chiao Tung University, Hsinchu, Taiwan. All subjects gave written informed consent by the laws of the country and the Research Ethics Committee. This study was approved by the Research Ethics Committee of National Chiao Tung University, Hsinchu, Taiwan.

### 2.2. Experiment Design

In this work, we designed a novel experimental scenario with combined SSVEP, working memory, and distraction tasks to recognize the BCI user’s mental focus state and lost-in-thought state, as shown in Figure 1. In previous SSVEP-BCI studies, subjects were looking directly at the selected SSVEP flicker frequency, and sustained use of the SSVEP flicker frequency caused fatigue or stress. In addition, we do not know that the participants were really focused on the SSVEP flashing frequency or they think about something else (i.e., lost-in-thought or mind wandering). The lost-in-thought state affects the performance of the BCI technologies. To address this problem, we designed a new experiment with a combination of SSVEP, work memory, and distraction tasks. To understand the procedures of the experimental scenario, each participant made ten practice trials. All participants completed the task twice, using SSVEP at 12 and then at 30 Hz. In each session, participants were asked to see the target in center of the computer screen. In this experimental scenario, we used 30 memory images (3 memory images × 10 batches), two SSVEP flashing frequencies at 12 Hz and 30 Hz, and distraction letter, to observe the effect of low and high flashing frequency in participants. Each participant performed the experiment carefully in each frequency. Each participant performed only one experimental task combined with SSVP, working memory, and distraction task. The total duration of each trial was 4 s, including the baseline or resting (1 s), the memory image, and the distraction target letters (1 s). In distraction task, the target letter appeared multiple times each time a memory image appeared. The SSVEP frequencies appeared as background for 3 s.

In this experiment, SSVEP flashing frequencies (12 Hz/30 Hz) stimuli were presented on a 27-inch monitor (ASUS VG278 H, 1920 × 1080 pixels). The distance between a participant and the monitor was fixed 60 cm. The working memory and the distraction tasks were designed to measure participants’ mental focus state in frontal lobe of the brain. The both 12 Hz and 30 Hz flashing frequencies were used to induce the SSVEP stimulation in occipital lobe of the brain. In this experiment, all participants first performed 12 Hz SSVEP flicker frequency and then 30 Hz flicker frequency. We used this spatial experimental scenario to recognize that the subject is in mental focus state or lost-in-thought state. In this study, we investigated neural activities of mental focus and lost-in-thought states in the occipital lobe and frontal lobe of the brain. Each participant performed task according to the experimental settings, as shown in Figure 1 and Figure 2.

### 2.3. Working Memory Task

In this experiment, we use the n-back working memory task to investigate the neural activities of mental focus and lost-in-thought states, as shown in Figure 2. We kept the task as simple as possible, so a participant was not involved under too much mental stress, but kept its difficulty at a necessary level to ensure that each participant has to pay attention to the images in the working memory [41]. The images used in the memory task were objects that are commonly found in everyday life. This ensured that all participants could recognize them easily. Thirty different types of scissors, toys, cups, and teapots were included, as shown in Figure 2. In this experiment, all images in the memory task were taken from the previous study of working memory [42]. The size of all images was 200 × 200 pixels and 2.4–2.6 degrees at a visual angle, which allowed the participants to identify the images and fix their eyes on them. According to the previous study of the work memory task [19], each image of the working memory task was presented randomly in the center of the computer screen with a black background for 1 s. After the presentation of three working memory images in a batch, the screen showed a message for each participant to prepare an answer for the question image, as shown in Figure 2. Each participant was instructed if the answer is correct, then press the right control key for “yes” and if the answer is incorrect, press the left control key for “no” on the keyboard.

### 2.4. SSVEP Flashing Frequencies at 12 Hz and 30 Hz

In this study, we used a low SSVEP flicker frequency at 12 Hz and a high SSVEP flicker frequency at 30 Hz, to investigate the effect of SSVEP stimulation in the occipital lobe of the brain. We tested whether SSVEP stimulation is stable at high flicker frequency (30 Hz). When the SSVEP flicker frequency increases, the amplitude of SSVEP decreases, but the background EEG does not decrease so quickly, therefore, the background EEG cannot be easily affected by the high flicker frequency of SSVEP. While the SSVEP flicker frequency drops, the amplitude of SSVEP increases so rapidly in the occipital lobe of the brain [43]. The high flicker frequency of SSVEP reduced the subjective discomfort of BCI users. For this reason, we use two SSVEP flashing frequencies at 12 Hz and 30 Hz in this study. The flashing frequency of SSVEP appeared on the screen for 3 s to induce a stable stimulation of SSVEP in the occipital lobe of the brain, as shown in Figure 1.

### 2.5. Distraction Task

In this study, Figure 1 shows the distraction task which was designed to investigate the mental focus state and lost-in-thought state. In this task, each participant waited for the appearance of a letter, which we called a target letter. Once the subject saw the target letter, they pressed the right control key to start the distraction task. In this distraction task, the target letter appeared several times, according to each time a memory image was presented, as shown in Figure 1. Eight capital letters were randomly selected as a source of distraction. The distraction task was lasted 1 s, as displayed in Figure 1. Each letter appeared on the screen for 250 ms, only one letter appeared on the screen at a time (due to persistent vision, participants may feel several letters blinking on the screen). The letters appeared randomly around the edges of a circle centered in the center of the screen and with a circumference of 450 pixels (5.4 to 5.9 degrees). The circumference was obtained from a pilot study, in which we presented a sequence of English letters at various distances from the center of each frame. The ideal distance was determined when the participants can identify the letters during fixing their eyes in the center of the screen. This process ensured that all letters were presented within the participants’ visual screen.

### 2.6. Acquisition of EEG Signals

In this work, Figure 3 displays the acquisition and preprocessing of the acquired EEG signals. All EEG signals were collected from all healthy participants’ using a Scan NuAmps Express system (Compumedics USA Inc., Charlotte, NC) under mental focus and lost-in-thought state. Thirty-two electrodes with a ground electrode and A1 and A2 reference electrodes were arranged according to the 10–20 international system. The EEG signals were imported into MATLAB R2017 b (The MathWorks Inc., Natick, MA, USA) using the EEGLAB toolbox (10.2.2.4bVersion, UC San Diego, Swartz Center for Computational Neuroscience, La Jolla, CA, USA) [35,36]. The EEG signals were recorded with a sampling rate of 500 Hz. All muscles artifacts, eye blink artifacts, and environmental artifacts were removed manually from all EEG signals [44,45]. In addition, to remove high frequency noise, EEG signals were down-sampled at 250 Hz and filtered using an FIR filter (1–50 Hz) in EEGLAB [35]. After filtering the EEG data, the epochs were extracted from the EEG data with a period of 4 s (−1 to 3 s) regarding the onset of each trial. The baseline (1 s), memory image (1 s) and the SSVEP frequency appeared for 3 s with the work memory image. The period of 1 s prior to onset was considered as the baseline for each trial. According to the performance of each participant in working memory and distraction tasks, we identified each participant cognitive state, such as mental focus or lost-in-thought. When a participant correctly answered (yes) to the working memory image in the distraction task, the corresponding EEG signals were labeled as a mental focus state. If a participant was unable to recognize a memory image and made an incorrect response (no), the corresponding EEG signals were labeled as a lost-in-thought state. Accordingly, we divided all EEG signals into two groups: (I) mental focus state and (II) lost-in-thought state. In this study, each participant performed a total of 72 trials for SSVEP flashing frequency at 30 Hz, and 72 trials for 12 Hz SSVEP flashing frequency. The fast Fourier transform (FFT) was measured using the MATLAB and EEGLAB function with EEG signals of mental focus and lost-in-thought state. The EEG dataset of all participants was merged for group analysis through MATLAB and EEGLAB function. In the final step, we measured the averaged event-related spectral perturbation (ERSP) using MATLAB with EEGLAB toolbox [36]. In addition, we measured the area under curve (AUC) through *trapz* function in MATLAB 2017b in the frequency range (1 to 40 Hz). We investigated the AUCs and power spectra of EEG signals from 0 s–3 s.

### 2.7. Event-Related Spectral Perturbation (ERSP) Analysis

The event-related spectral perturbation (ERSP) are gradually used in the EEG signal analysis to see mean event-related changes in spectral power over time in a wide range of frequencies. It is measured by event-related desynchronization (ERD) and synchronization (ERS) [36]. For calculating an ERSP needs computing the power spectrum in a sliding latency window then averaging across EEG data trials. The color at each image pixel then shows power (in dB) at given frequency and latency relative to the time locking event. Usually, for *n* trials, if $f_{k}\left(f,t\right)$, is the spectral estimate of trial *k*at frequency *f* and time *t*.
$$ERSP\left(f,t\right)=\frac{1}{n}\overset{n}{\underset{k=1}{\sum }}{\left|f_{k}\left(f,t\right)\right|}^{2}$$


### 2.8. Statistical Analysis

In this study, we investigated the neural activities of mental focus and lost-in-thought states. The statistically significant difference (*p* < 0.05) between the EEG neural activities of mental focus state and lost-in-thought state in the time and frequency domains were measured using the bootstrap significant test in the EEGLAB toolbox [36,46]. In addition, we investigated multiple comparisons, significance values were corrected using the FDR method [47] in EEGLAB toolbox [36]. We also performed pairwise significance (*p* < 0.05) *t*-tests in the power spectral density (PSD) and area under curve (AUC) analysis in frequency (1–40 Hz) between mental focus state and lost-in-thought state.

### 2.9. BCI User’s Mental State Changes Monitoring Classification System

In addition, in this work, we designed a classification system to detect BCI users’ cognitive state changes based on their SSVEPs. In this study, we applied classification algorithms to investigate the relationship between mental focus and lost-in-thought states at 12 Hz and 30 Hz SSVEPs. We tried two classifiers that are commonly used in EEG signal classification including k-nearest-neighbor classifier (KNNC) and Bayesian network [48]. KNNC is a simple classifier. However, it is sensitive to the curse of dimensionality, former study shown that KNN is effective with low dimensional feature vectors [48]. Based on these features, we observed the performance of the KNNC (K = 1) classifier in this study. Bayesian network classifier combines directed acyclic graphs with Bayesian probability. Rezaei et al. [49] mentioned that the Bayesian network has good accuracy and more reliable classification on their EEG data sets than neural network. We collected training and testing data from the EEG data in channels O1 and O2, because these channels were most connected to the SSVEP (visual region) of the brain. To avoid bias in the performance evaluation, each subject’s EEG data were divided into a training set and a testing set. All EEG signals were preprocessed and the PSD of the SSVEP was measured. Each subject had an average 42 mental focus state trials and 30 lost-in-thought state trials. The training and testing trials were designated randomly. To prevent the over-fitting problem and increase the generalizability of the trained classifier. The EEG data set was divided into 10 subsets. Each time, one of the 10 subsets was used as the test set and the other 9 subsets were put together to form a training set. Then the average error across all 10 times was computed. That is division of data into a training set (90% data) and a test set (10% data) was performed ten times. At each time, training and testing processes were performed.

## 3. Results

In this work, Figure 4 shows the average event-related spectral perturbation (ERSP) across the participants during mental focus and lost-in-thought states. In this study, we investigated average neural activities of mental focus and lost-in-thought states in frontal (F3, Fz, F4) lobe and occipital (O1, Oz, O2) lobe of the brain. Because we want to observe the neural activities of the entire frontal and occipital cortices under mental focus and lost-in-thought states. Previous study reported that the frontal and occipital cortices were selected as a region of interest because the frontal lobe has been associated with the executive function, working memory, mental focus state, attention, and lost-in-thought state [50], and the occipital lobe has been linked to visual stimulation and SSVEP-BCI task [13,23,51]. Therefore, these two brain regions were selected to explore the neural activities of mental focus and lost-in-thought states.

### 3.1. Neural Activities of Mental Focus and Lost-in-Thought States in the Frontal Lobe

Figure 4A–D shows the brain activities changes during mental focus and lost-in-thought states at 12 Hz SSVEP frequency. Figure 4A displays the location of the EEG channels (F3, Fz, F4) in the frontal lobe of the brain. Figure 4B,C shows the statistically significant (*p* < 0.05) higher powers of the delta band (1–4 Hz) and theta band (4–7 Hz) during the period of 1 to 1000 ms in mental focus state than in the lost-in-thought state at frontal lobe of the brain. In addition, we observed a decrease alpha band (8–12 Hz) power (1 to 1000 ms) in lost-in-thought state than in the mental focus state at the frontal lobe. Figure 4B,C shows statistically significant (*p* < 0.05) increased beta band (13–30 Hz) power from 1000 to 3000 ms, and a decrease beta band power from 1 to 1000 ms was found in the mental focus state than in the lost-in-thought state at the frontal lobe. Moreover, Figure 4D shows the area under curve (AUC) of power spectral changes during mental focus and lost-in-thought states in the frontal lobe. The AUC was measured in the 1–40 Hz frequency range between powers spectral of mental focus and lost-in-thought states. We found AUC statistically significantly (*p* < 0.05) greater in the mental focus state than in the lost-in-thought state, as shown in Figure 4D. These neural activities revealed that each subject was in the mentally focused state during SSVEP and working memory task. These findings show that the frontal lobe was functionally activated in mental focus state than in the lost-in-thought state.

Figure 5A–D displays the neural activities during mental focus and lost-in-thought states at 30 Hz SSVEP flashing frequency. Figure 5A presents the location of the EEG channels (F3, Fz, F4) in the frontal lobe of the brain. Figure 5B,C shows the statistically significant (*p* < 0.05) greater powers of the delta band (1–4 Hz) and theta band (4–7 Hz) during the period of 1 to 1000 ms in mental focus state than in the lost-in-thought state at frontal lobe. Additionally, we investigated a decrease alpha band (8–12 Hz) power (1–1000 ms) in lost-in-thought state than in the mental focus state at the frontal lobe. Figure 5B,C reveals statistically significantly (*p* < 0.05) increased beta band (13–30 Hz) power from 1000 to 3000 ms, and a decreased beta band power from 1 to 1000 ms was found in the mental focus state than in the lost-in-thought state at the frontal lobe. However, Figure 5D shows the AUC of power spectral changes during mental focus and lost-in-thought states in the frontal lobe. The AUC was calculated in the 1–40 Hz frequency range between powers spectral of mental focus and lost-in-thought states. We observed AUC statistically significantly (*p* < 0.05) higher in the mental focus state than in the lost-in-thought state, as shown in Figure 5D. These neural markers show that each subject was in the focused state during SSVEP and working memory task. These results show that the frontal lobe was functionally stimulated in mental focus state than in the lost-in-thought state.

### 3.2. Neural Activities of Mental Focus and Lost-in-Thought States in the Occipital Lobe

Figure 6A–D shows the neural mechanism of mental focus and lost-in-thought states at an SSVEP flicker frequency of 12 Hz. Figure 6A displays the location of the EEG channels (O1, Oz, O2) in the occipital lobe of the brain. Figure 6B,C shows statistically significantly (*p* < 0.05) increased power of the delta band (1–4 Hz) from 1 to 3000 ms and decreased power of alpha-beta bands from 1 to 1000 ms in the lost-in-thought state than in the mental focus state at the occipital lobe. These findings show that the participants were lost-in-thought state or mind wandering during SSVEP and working memory task. Additionally, Figure 6B,C shows statistically significantly (*p* < 0.05) increased power of alpha-band from 1 to 3000 ms and beta band power from 1000 to 3000 ms in the mental focus state than in the lost-in-thought state at the occipital lobe. Moreover, Figure 6D displays the significantly (*p* < 0.05) greater AUC in the mental focus state compared to the lost-in-thought state. These results show the stimulation of SSVEP at 12 Hz flicker frequency in the occipital lobe.

Figure 7A–D shows the neural activities of mental focus and lost-in-thought states at an SSVEP flicker frequency of 30 Hz. Figure 7A shows the location of the EEG channels (O1, Oz, O2) in the occipital lobe of the brain. Figure 7B,C displays statistically significantly (*p* < 0.05) increased power of the delta band from 1 to 3000 ms, and decreased power of alpha-beta bands from 1 to 1000 ms in the lost-in-thought state than in the mental focus state at the occipital lobe. These results reveal that the participants were lost-in-thought state or mind wandering state in SSVEP and working memory task. Furthermore, Figure 7B,C shows the statistically significantly (*p* < 0.05) increased power of alpha-band from 1 to 3000 ms and beta band power from 1000 to 3000 ms in the mental focus state than in the lost-in-thought state at the occipital lobe. Likewise, Figure 7D shows the statistically significantly (*p* < 0.05) higher AUC in the mental focus state than in the lost-in-thought state. These neural markers show the stimulation of SSVEP at 30 Hz flicker frequency in the occipital lobe of the brain. Additionally, we observed that the 30 Hz SSVEP flashing frequency was more comfortable for participants than the 12 Hz SSVEP frequency. These EEG results show that the 12 Hz flashing frequency induced the greater SSVEP power stimulation than in the frequency of 30 Hz in the frontal and occipital cortices.

### 3.3. Power Spectral Density (PSD) during Mental Focus and Lost in Thought States

Moreover, we investigated power spectral density between the mental focus and lost-in-thought states at 12 Hz and 30 Hz SSVEP powers. Figure 4C and Figure 5C show the paired *t*-test, statistically significant (*p* < 0.05) differences in PSD of delta, and the band powers in the frontal lobe of the brain. The PSD of delta and theta band powers were observed significantly higher in mental focus state than in lost-in-thought state in the frontal lobe. In addition, Figure 6C and Figure 7C display the slight significant differences in PSD of delta, alpha, and beta band powers in the occipital lobe. The PSD of the delta, alpha, and beta band powers were investigated significantly higher in mental focus state compared to the lost-in-thought state at the occipital lobe. These results show the neural markers of mental focus and lost-in-thought states.

### 3.4. Mental State Changes Monitoring Classification Results

The machine learning algorithms (k-KNNC) and Bayesian network [48] were applied to classify the EEG signals of mental focus and lost-in-thought states at 12 Hz and 30 Hz SSVEPs over the occipital lobe (i.e., channels O1 and O2). The power spectral density (PSD) of mental focus and lost-in-thought states was used as an input feature for the machine learning algorithms. To avoid bias in the performance evaluation, each participants EEG datasets were divided into a training set and a testing set. The training and testing datasets were selected randomly. The EEG data set was divided into 10 subsets. Each time, one of the 10 subsets was utilized as the testing set and the other 9 subsets were put together to form a training set. Then the average error across all 10 times was computed. That is division of data into a training set (90% data) and a test set (10% data) was performed ten times. At each time, training and testing processes were achieved. 

Figure 8 displays the performance of Bayesian network and KNN classifiers. All classification results were related to the testing phase. The classification performance rates were the average of 10 times of testing obtained in the 10-fold cross validation. In this work, for 30 Hz, we achieved the best classification accuracy of 97% for single subject S10 with the Bayesian network classifier. This revealed that 30 Hz high-frequency SSVEP can be utilized to monitor the mental focus state of a single subject. In addition, for 12 Hz, 94% for single subject S13 with the KNN and the Bayesian network reached the maximum classification accuracy. However, the average classification performance across participants for the two types of classifiers were 77% to 80% at 12 Hz SSVEP frequency and 77% to 79% at 30 Hz SSVEP frequency. These findings show cognitive state changes in mental focus or lost-in-thought state affect the performance of the BCI users in the sustained use of SSVEP.

In this study, we have presented intra-subject and inter-subject variability, as shown in Figure 8. Results from S5, S8, and S13 show that changes in the neurophysiological states (mental focus or wandering) of the individual subject affect the performance of the SSVEP-BCI system.

## 4. Discussion

In this study, we investigated the neural activities of mental focus and lost-in-thought states. First time, we designed a novel experimental scenario with combined SSVEP, working memory and distraction task, to recognize the neural markers of mental focus state and lost-in-thought states. We investigated more increased power of the delta, theta, and beta bands in the mental focus state than in the lost-in-thought state at the frontal lobe of the brain. In addition, we observed more increased power of the delta, alpha, and beta bands in the mental focus state than in the lost-in-thought state in the occipital lobe. The increased power of the delta, theta bands in the frontal lobe of the brain are directly related to the mental focus state of the participants. Similar findings have been investigated in previous EEG studies of visual attention and working memory task in frontal lobe [24,28,50].

### 4.1. The EEG Power of the Delta (δ), Theta (θ, Beta (β) Bands Increased in the Frontal Lobe during Mental Focus State

The preceding study reported that the frontal lobe of the brain plays an important role in sustained attention [22]. In addition, previous study described that the frontal lobe is associated with maintenance and development of working memory [23]. Accordingly, we investigated the increased delta (δ), theta (θ), and beta (β) band powers in the frontal lobe during mental focus state (i.e., attention state), as shown in Figure 4 and Figure 5. Evidently, this type of increased theta band power in the frontal lobe has been described in numerous studies on working memory tasks [24,28,52,53]. In our study, the increased power of the theta band in the frontal lobe of the brain shows that the participants were mentally focused on working memory images in the distraction task with SSVEP target frequencies at 12 Hz and 30 Hz. Additionally, we observed that higher beta band power increased over the frontal lobe of the brain in mental focus state compared to the lost-in-thought state (1000 to 3000 ms), as shown in Figure 4 and Figure 5. This change in EEG activity was related to SSVEP oscillations at frontal lobe. The beta band power was decreased in the frontal lobe of the brain during the period from 1 to 1000 ms. This EEG activity change was related to working memory task, as shown in Figure 4 and Figure 5. The study of M. Bauer et al. have also investigated similar EEG neural activities during a visual attention task [25]. A former study of EEG signals reported that the decrease in the power of the beta band has been related to human attention or mental focus state [25], this finding also suggests that the frontal lobe was directly connected with attention or mental focus state [24,26]. Moreover, in this work we found a common beta band power increased in the frontal lobe during mental focus and lost-in-thought states from 1000 to 3000 ms. This EEG activity changes revealed the oscillation of SSVEP target frequencies at 12 Hz and 30 Hz in the frontal lobe of the brain. Furthermore, our study show that the power of the delta band increased over the frontal lobe during lost-in-thought or mind wandering state. This finding is consistent with the previous study of the mind wandering [26,54]. This EEG neural marker is related to the decrease of the alertness in the BCI users or inattention of the participants [29,30,31,32,33,34,54]. Additionally, the power of the alpha band was found suppressed in the frontal lobe of the brain during the lost-in-thought state, as shown in Figure 4B and Figure 5B. These findings also suggests that participants (BCI users) were under lost-in-thought or mind wandering state during working memory task [26]. Finally, in the frontal lobe, we found that the higher SSVEP power changes in the mental focus state than in the lost-in-thought state at 12 Hz and 30 Hz target frequencies [55].

### 4.2. The EEG Power of the Delta (δ), Alpha (α), Beta (β) Bands Increased in the Occipital Lobe at Mental Focus State 

In SSVEP-BCI study of I. Kramberger et al. have shown that the occipital lobe of the brain plays an important role in BCI technologies [56]. The former studies of the working memory task reported that the increased beta band power in the parietal and occipital lobes [23,28]. Accordingly, in our study, we also observed the increased delta (δ), alpha (α), and beta (β) band powers during mental focus state in the occipital lobe of the brain, as shown in Figure 6 and Figure 7. Moreover, we found that the higher beta band power increased over the occipital lobe of the brain in mental focus state than in lost-in-thought state from 1000 to 3000 ms (i.e., related to SSVEP stimulation), as shown in Figure 6B and Figure 7B. The beta band power investigated was decreased in the occipital lobe of the brain during the period from 1 to 1000 ms in mental focus state. These EEG neural markers have been related to previous study of visual attention [25]. In addition, we observed that the EEG activity of the beta band power decreased in occipital lobe, it was related with mental focus state or attention [25], this finding shows that the occipital lobe was directly related with mental focus state or visual attention [24]. The EEG power of delta band was found increased in the occipital lobe during lost-in-thought state. This neural marker is consistent with the preceding study of mind wandering [26,54]. Also, this EEG neural marker was associated with the decrease of alertness or inattention of the participants in working memory task [54]. Moreover, preceding EEG studies reported that the delta band power increased in occipital lobe of the brain is associated with a decrease in the level of human alertness [29,30,31,32,33,34], respectively. These changes in EEG activity revealed the neural markers of mental focus state and lost-in-thought state in the occipital lobe of the brain. These novel EEG neural markers can be used in BCI technologies to recognize the human mental focus and lost-in-thought states.

### 4.3. Application and Limitation of This Study

The application of the present study is in development of BCI technologies to recognize the mental focus and lost-in-thought states of participants, these states affect the performance of the BCI system [56,57,58,59]. For example, if the subject is in lost-in-thought state according to his/her EEG brain dynamics or neural markers, the adaptive hybrid BCI model of lost-in-thought state may change in mental focus state. Accordingly, these findings are useful to improve the performance of the current BCI technologies. Another application of the current study is in digital marketing and web advertising. Usually, web advertising is a hot research topic for their emerging value in digital marketplace. According to the EEG neural markers of humans in a mental focus state and lost-in-thought state, we can easily recognize that people are really focusing on web advertising or they are in mind wandering state or lost-in-thought state [60,61,62,63]. 

In addition, we present the limitations of the current study, for example, the SSVEP flashing stimuli, the working memory images, and the distraction letters were used in 2-dimensional shapes. Future work can develop a more realistic scenario in the virtual reality (VR) system and augmented reality (AR) system to enhance the future impact of this study. In this study, we used a total of thirteen participants, it is a small population size. Young healthy participants joined this experiment. The level of attention (mental focus) was detected by the correct response for each image in the working memory tasks. We found no neural EEG markers related to mental fatigue in the frontal and occipital lobes of the brain. In future work, we will explore all of these limitations.

## 5. Conclusions

In this study, we developed a novel experimental scenario to recognize the mental focus and lost-in-thought state of BCI users. In BCI technologies, the participants were focused on the SSVEP flashing frequency, but after a while, participants may have got fatigued, stressed, lost-in-thought, or in a state of mind wandering during sustained use of SSVEP; these cognitive states affect the BCI system performance. Therefore, in this work, we combined SSVEP, working memory, and distraction tasks, to investigate the neural activities of mental focus and lost-in-thought states in the occipital lobe and the frontal lobe of the brain. We observed significant SSVEP power modulation during the neural activities of mental focus and lost-in-thought in frontal and occipital lobes. However, SSVEP power was found increased in mental focus state than in lost-in-thought at frontal and occipital lobes. In addition, the average classification performance rate across participants for the Bayesian network classifier was achieved 80% in mental focus state. In addition, we achieved the best classification accuracy of 97% for individual subject S10. These outcomes show how cognitive state changes (mental focus or lost-in-thought) affect the performance of the BCI user. These novel EEG neural markers can be applied in BCI technologies to recognize the user’s mental states.

## Figures and Tables

**Figure 1 sensors-20-03169-f001:**
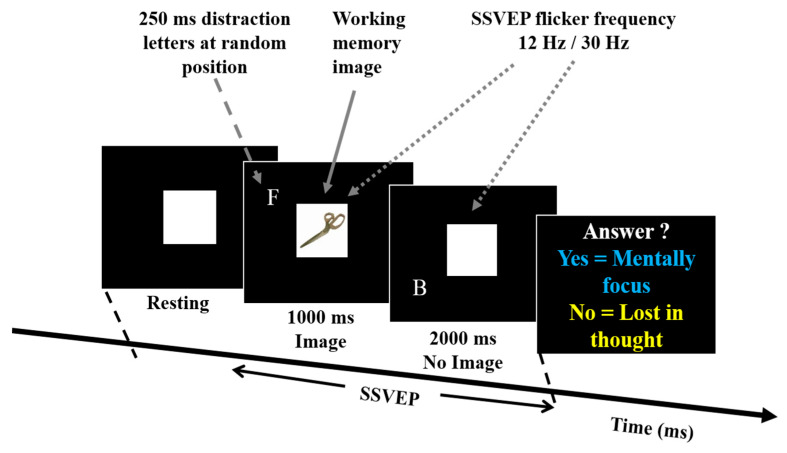
Experimental setup used for steady-state visual evoked potential (SSVEP), working memory, and distraction tasks. In this experiment working memory images, SSVEP flashing frequencies at 12 Hz and 30 Hz, and distraction letters were used as stimuli. In this scenario, all participants were instructed to look at the center of the screen and perform SSVEP-BCI task. This experiment was designed to recognize the brain dynamics of mental focus and lost-in-thought states.

**Figure 2 sensors-20-03169-f002:**
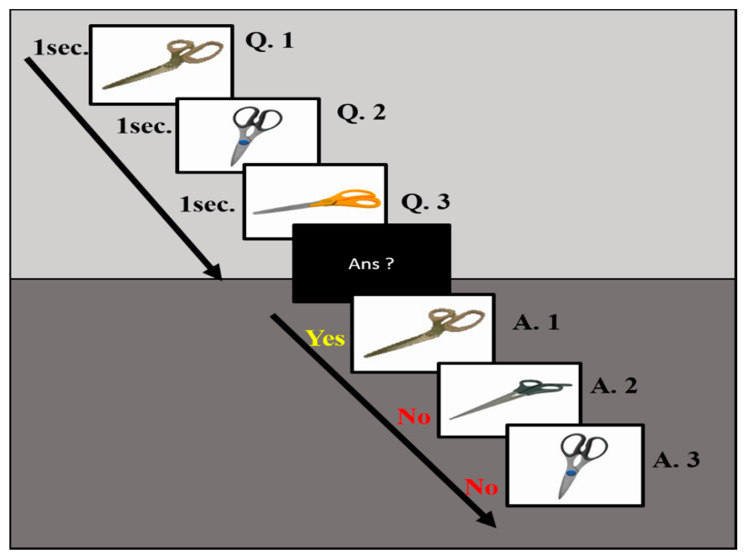
The demonstration of working memory images was used to recognize the state of human mental focus. In the working memory task, a sequence of images was presented one by one on a computer screen. The duration of each image appeared was 1 s. Each subject was instructed to memorize images in the correct order. The three images were presented as question memory images, three other images were used as the response image. This sequence was presented repeatedly. All participants were instructed to identify the correct image in the work memory task.

**Figure 3 sensors-20-03169-f003:**
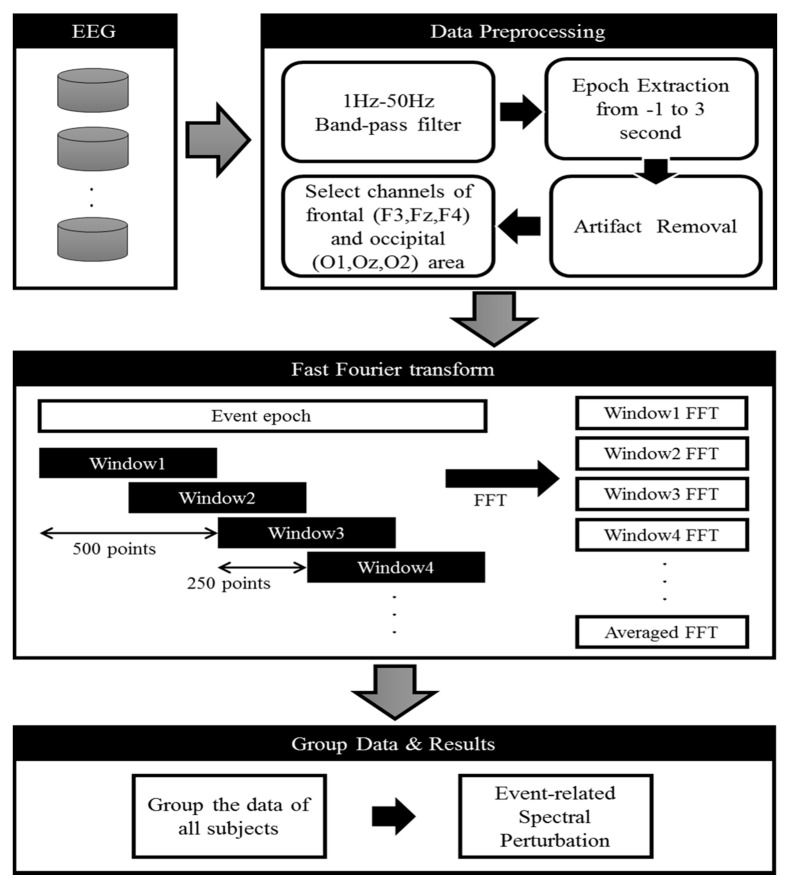
Flowchart of EEG analysis during mental focus and lost-in-thought states.

**Figure 4 sensors-20-03169-f004:**
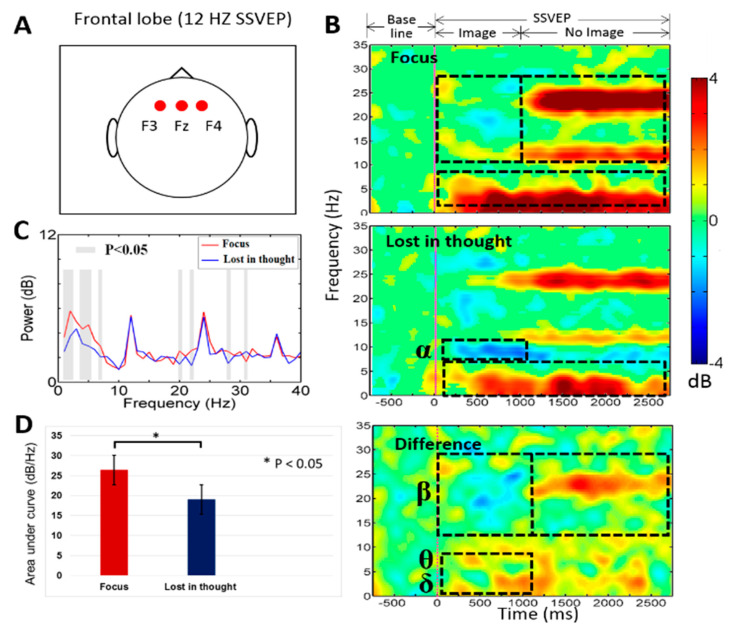
(**A**) Displays electrodes (F3, Fz, F4) location in frontal lobe. (**B**) The event-related spectral perturbation (ERSP) was observed during mental focus and lost-in-thought states at 12 Hz SSVEP flicker frequency. The pink vertical line shows the stimulus onset. Statistic at *p* < 0.05. Color bar indicate the scale of significant ERSP in decibel (dB). (**C**) The power spectral density (PSD) for the mental focus state (red line) and lost-in-thought state (blue line). The grey areas represents statistically pairwise significant difference (*p* < 0.05) in *t*-test between mental focus and lost-in-thought states. (**D**) The area under curve (AUC) was measured in mental focus and lost-in-thought states (* *p* < 0.05).

**Figure 5 sensors-20-03169-f005:**
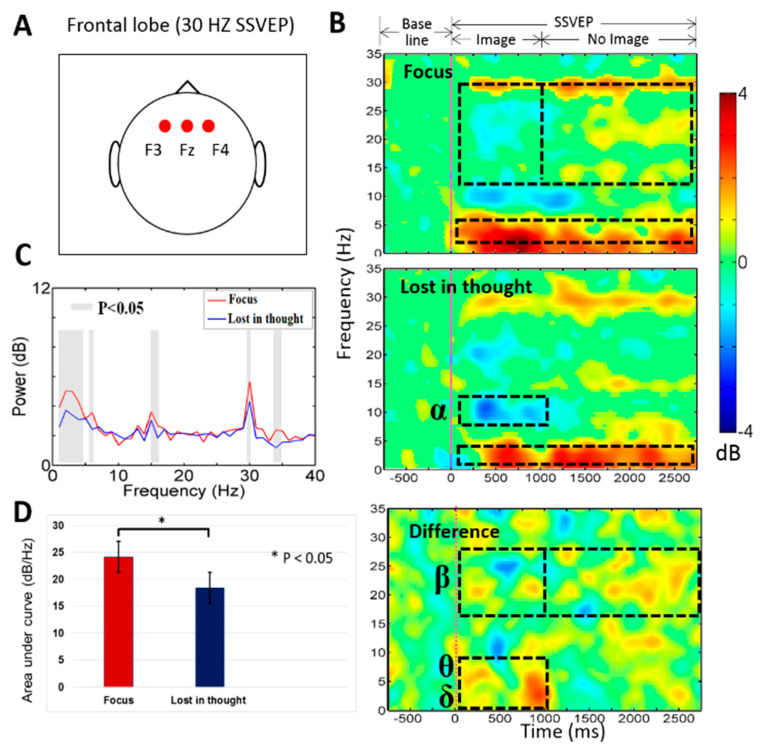
(**A**) Presents F3, Fz, F4 electrodes location in frontal lobe. (**B**) The ERSP was observed during focus and lost-in-thought states at 30 Hz SSVEP. The pink vertical line display the stimulus onset. Statistic at *p* < 0.05. Color bar indicate the scale of significant ERSP in decibel (dB). (**C**) The PSD for the mental focus state (red line) and lost-in-thought state (blue line). The grey areas represents the statistically pairwise significant difference (*p* < 0.05) in *t*-tests between focus and lost-in-thought states. (**D**) The AUC was investigated in mental focus and lost-in-thought states (* *p* < 0.05).

**Figure 6 sensors-20-03169-f006:**
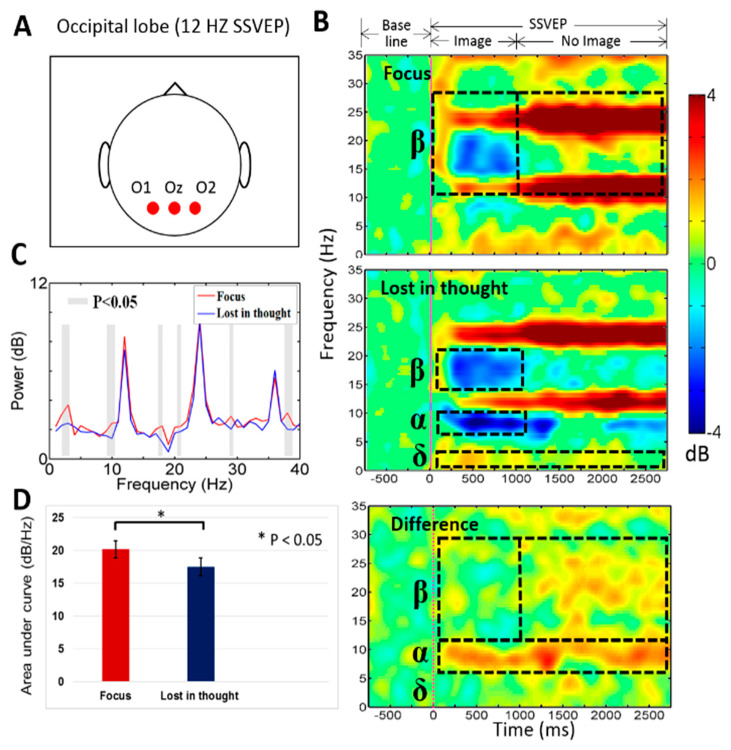
(**A**) Shows O1, Oz, O2 electrodes location in occipital lobe. (**B**) The ERSP was investigated during mental focus and lost-in-thought states at 12 Hz SSVEP. The pink vertical line display the stimulus onset. Statistically significant at *p* < 0.05. Color bar indicate the scale of significant ERSP in decibel (dB). (**C**) The PSD for the mental focus state (red line) and lost-in-thought state (blue line). The grey areas presents statistically pairwise significant difference (*p* < 0.05) in *t*-tests between the mental focus and lost-in-thought states. (**D**) The AUC was measured in mental focus and lost-in-thought states (* *p* < 0.05).

**Figure 7 sensors-20-03169-f007:**
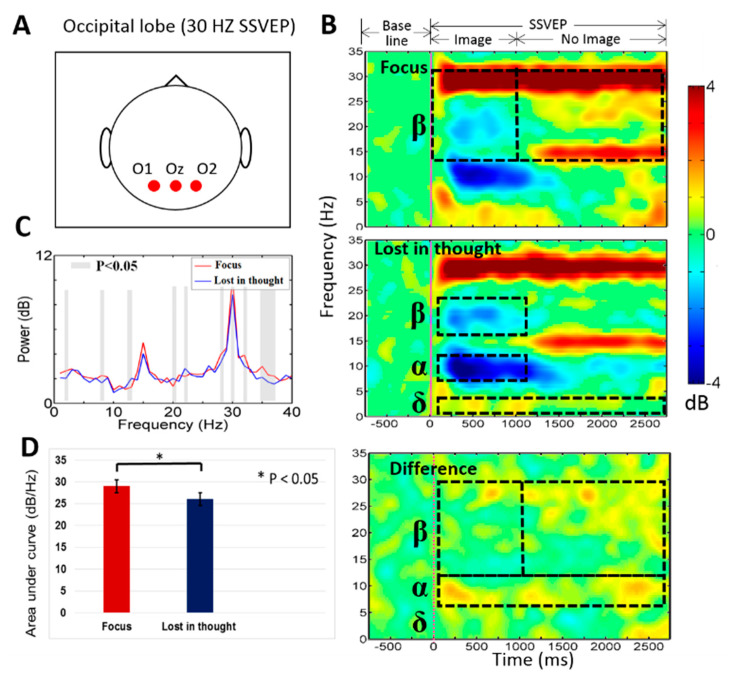
(**A**) Shows O1, Oz, O2 electrodes positions in occipital lobe. (**B**) The ERSP was investigated in mental focus and lost-in-thought states at 30 Hz SSVEP. The pink vertical line show the stimulus onset. Statistically significant at *p* < 0.05. Color bar indicate the scale of significant ERSP in decibel (dB). (**C**) The PSD for the mental focus state (red line) and lost-in-thought state (blue line). The grey areas represent statistically pairwise significant difference (*p* < 0.05) in *t*-tests between the mental focus and lost-in-thought states. (**D**) The AUC was measured in mental focus and lost-in-thought states (* *p* < 0.05).

**Figure 8 sensors-20-03169-f008:**
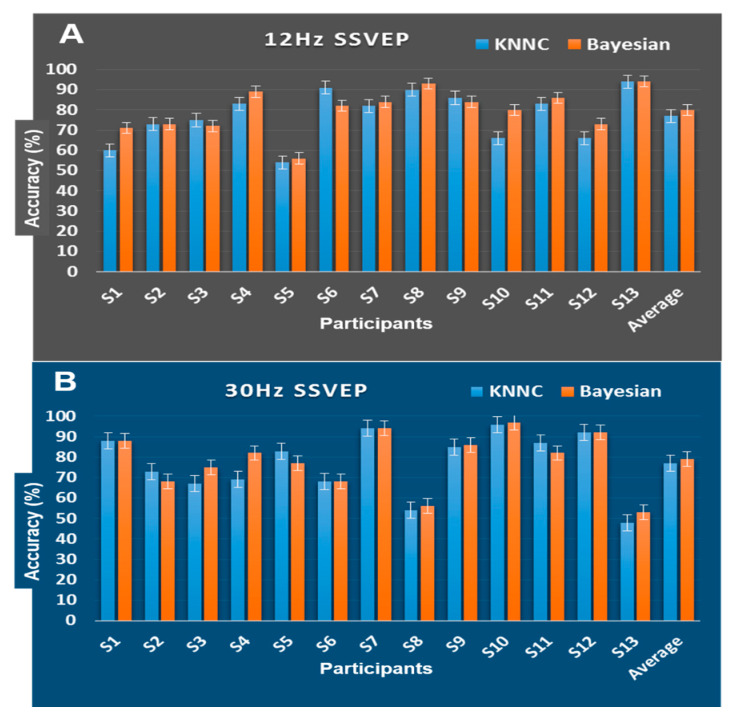
The classification outcomes of the Bayesian network and KNN classifier with SSVEP of 12 Hz and 30 Hz at O1 and O2 (occipital lobe). (**A**) The accuracy (%) of the Bayesian network and KNN classifier using SSVEP of 12 Hz at O1 and O2. (**B**) The accuracy (%) of the Bayesian network and KNN classifier with SSVEP of 30 Hz at O1 and O2.

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
