# Peer review of "Exploration of User’s Mental State Changes during Performing Brain–Computer Interface"

_sensors, 2020, doi:10.3390/s20113169_

Round 1
Reviewer 1 Report
The paper presents a study to determine de influence of the attentional level (focused or lost in thought) on the spectral power of the EEG-based Steady-state Visual Evoked Potentials (SSVEPs). For that purpose, the authors introduce an experimental paradigm with SSVEPs, and working memory and distraction tasks. The results showed that the attentional levels targeted did have an influence on the power of the SSVEPs frequencies, both at frontal and occipital areas. The authors also reported automatic classification results of the attentional level based on the spectral differences found with two types of classifiers, reaching an average of 80% accuracy per subject.
There are several concerns and issues that prevents the paper from being acceptable as it is:
- The authors should provide a grounded rationale for the hypotheses they propose, described in the last paragraph of the Introduction section (lines 103-108): Why did they think that the SSVEPs frequency would increase at frontal sites given that previous studies reported the contrary? Why did they think that SSVEPs at 30Hz would be more comfortable for the subjects?
- The authors highlights in the Introduction the interest and benefits of the BCI applications for people with neural and movement disorders, which are at the very core of the BCI’s sense of existence. However, they tested their hypotheses in healthy young subjects. This carries to problems. On the one hand, the damaged brains will not likely show the same brain activity patters than the healthy brains. On the other hand, the brain is still rewiring and developing until 25 year old, specially the frontal lobe (this is widely evidenced), which is one of the target areas of the study. The subjects in the present study are all between 18 and 25 years old.
- The population size is really small (N=13). It might certainly provide a proof of concept, but still the contribution is reduced.
- The experimental protocol description is highly confusing and controversial. I was not able to fully understand it even after reading it several times:
- Did the subjects perform two tasks, one WM related and other distraction related? I guess that it is only one task because the results section only report outcomes from one task, but it is not clear at all. If it was one task only, how the WM and distraction tasks integrated in the protocol?
- Caption of Figure 2 and text states that the images are presented during one second. However, in the Figure 2 graphics shows 4s for each image.
- In the distraction part, the subject is presented with the target letter first, and then with random letters. Did the subjects have to answer whether the target letter appeared any time among the random letters?
- The authors state that subjects performed several practice tests before the actual trials. How many practice tests they perform of each type?
- Each subject performed the protocol for 12Hz and 30Hz SSVEPs. In what order? Always 12Hz first and then 30Hz? All the subject in the same order? A frame effect might have happened.
- The two last issues carry an implicit question. Did the authors control the mental fatigue? Mental fatigue is widely evidenced to affect EEG activity.
- There are also some concerns and missing information about the methodology of the analysis:
- Are the graphics in Figure 4, 5, 6 and 7 the average of the three electrodes? Why not consider each electrode separately?
- To what time interval does the power spectra and AUCs in Figures 4, 5, 6 and 7 correspond? 0s-3s? 1s-3s? The whole epoch? Why not split the analysis for the image and no image intervals? This would be much more informative for the study purposes.
- Why have not the results of frontal lobe at 30Hz been reported?
- Why did the authors include Oz if they later only used O2 and O1with the classifiers “because these channels were most connected to the SSVEP (visual region) of the brain? If in the previous analyses the authors were reporting the average of the three electrodes then Oz might be introducing noise.
- Why did the authors only use occipital features for the classifiers, given that also significant frontal differences were found and these would not interfere with the occipital SSVEPs features for BCI?
- Why did the authors select k=1 for the kNN classifier? Did the authors try other values of k?
- The authors should elaborate on the reasons for not to conduct the analyses in an individual basis and then integrate them, given that BCIs present a high intra-subject variability and the most accurate performance is reached when the features are individually customized.
- It is highly controversial to label trials as focused or lost in thought on the solely basis of the responses to the WM and distraction tasks. Why did not the authors ask the subjects whether they were lost in thoughts or not? Or any other type of subjective questionnaire? Moreover, it cannot be assured that the same level of attention is reached from the positive or negative answer of both tasks (WM and distraction). In fact, the distraction task might be biased to evoke more lost in thought states, since it forces the subjects to change the focal distance of their eyes in order to perceive the letter with the peripheral vision, and this change in the focal distance may induce a mind wandering state.
- With respect to the results, the authors stated that the Bayesian network classifier with 30Hz spectral features is the best approach for the automatic attentional level detection. However, given the results in Figure 8, the average results are slightly higher for the 12Hz SSVEP features for both classifiers, and also they present less subjects under 60% accuracy.
- The authors should also perform additional analyses or provide some explanation for the subjects with the worst result (S5, S8 and S13).
- The hypotheses proposed were just marginally satisfied. The authors should elaborate more on this in the discussion:
- The hypothesis about the higher comfort of subjects at 30Hz SSVEPs was not treated at all from the results.
- The hypothesis about the increased activity of the SSVEP frequency in the frontal areas was not confirmed.
- The limitations should be expanded: population size, healthy young subjects, way of assessing the attentional level, uncontrolled mental fatigue…
In summary, the authors must clearly define and ground their hypotheses, detail the methodology so that it can be replicated by anybody else from the content of the paper, argue why the methodology is appropriate to test the hypotheses, completely present the results of all experiments, discuss whether the results support the hypotheses introduced or not and how this contribute to the field (corroborate or contradict previous results or provide new knowledge) and clearly state the limitations of the study and how these affect the interpretation and contribution of the results.
Minors:
- Use carefully the term “highly activated” from EEG spectrum. More spectral power in a specific frequency band does not necessarily means more activation or more functional involvement.
- Language should be revised. There are some ungrammatical sentences or weird expressions.
Author Response
Dear Editor,
Please find the attached file for our response. We have revised the all statements of the manuscript carefully, according to reviewer suggestions. Thank you.

Reviewer 2 Report
Observations:
Clarify in section 2.1 if there are 13 or 16 participants..
Why not directly use a sampling rate of 250 Hz?
How the event-related spectral perturbation is defined?
The corresponding methodology is not described in section 3.4, for example how the patterns are formed, what the classes are and why these classifiers were used.
Author Response

(The authors gave the same response as above.)

Reviewer 3 Report
Introduction
The relevant literature is reviewed, and hypotheses are made, but the flow of this section is difficult to track. The justification for the purpose of this study and its hypotheses is therefore unclear. Here are a few recommendations to assist with this:
SSVEP is introduced early in the first paragraph of this section but it is not described properly until the second paragraph. To improve the flow of this introduction, you might want to consider adding an explanation for SSVEP here.
I’d also start a new paragraph where you introduce your study; from the line: “To investigate the brain dynamics of BCI users in mental focus and lost 53 in thought states, we designed a novel experimental scenario combined with SSVEP-BCI, working 54 memory and distraction tasks.” As you’ve concluded the introductory material concerning BCI technology here, and stated the problem under investigation, a new paragraph should be started to introduce the next discussion topic (i.e., the use of SSVEP in BCIs).
The end of the second paragraph is also rather confusing. “In this study, first time we modified and adopted this working memory task with SSVEP-BCI and distraction tasks, to investigate the neural activities of mental focus and lost in thought states.” Which study are you referring to? The preceding sentence doesn’t refer to any study so it’s unclear what this statement relates to.
The next two sentences are just as unclear: “In our study, subject’s responses in the working memory task was considered as the ground truth to verify whether a subject was in mental focus state or lost in thought state.” What does our study refer to? Can you clarify what kind of responses corresponded to being mentally focussed vs. lost in thought here? It’s unclear what ‘subject’s responses’ means otherwise.
“The previous EEG study reported that theta (4-7 Hz) band power increased in the frontal lobe of the brain under working memory task [3].” Here you’ve discussed the outcomes of a prior study but fail to describe key methodological details such as stimulation rate, memory task type, and temporal measurement of theta power. This is the first time the EEG is mentioned, however, it is not described until the final paragraph. These findings need context – firstly, what is theta? did theta band power increase in the frontal lobe during the working memory task overall, relative to a pre-task resting state? Or was there greater frontal theta power occurring when participants responded correctly in the task and therefore were mentally focussed? Please clarify when theta was measured.
In the third paragraph, the authors introduce their current study’s methodology before summarising other prior studies that are relevant to the present one’s design. This is the reverse order of what is typically written. Please review and discuss the relevant literature before introducing the current study and hypotheses. The current format creates confusion as you discuss the use of 12 and 30 Hz flicker frequencies before explaining why these frequencies were chosen.
Furthermore, there is insufficient detail concerning the EEG findings from studies utilising 12 and 30 Hz flickers. These should be reviewed so that specific hypotheses can be made. As it stands, the hypotheses are not justified by the content presented in the introduction.
A minor comment about the use of the word ‘subjects’. Good clinical practice and most fields of research recommend that this term no longer be used. Instead, terms like participants and volunteers should be used.
Methods
How were your participants recruited? Were they screened for other EEG altering factors like caffeine and drug use?
A sample size of 13 participants is incredibly small. How did you verify that you had enough power to statistically detect differences between mentally focussed vs. lost in thought states?
Can you clarify if the participants completed the task twice, using SSVEP at 12 and then 30 Hz, or was this done once, and the flashing rate varied between stimuli in the one task?
Were the images randomly presented to the participant? Was the SSVEP flash also fixed to appear between images? If the tasks were completed twice (i.e., once with a 12 Hz flicker and then using a 30 Hz flicker) – how did you control for possible familiarisation effects after the first task?
For replicability, could the authors state the EEG recording system and software that was used, as well as name the 32 electrodes used? Was there a ground electrode? How was the online data referenced? Were there any offline re-referencing procedures conducted? Was an online filter used to suppress mains interference during the recording? Did you remove the DC component from your epochs?
The epoching process is rather confusing in-text, whereas the figure demonstrates this quite well. Can the authors clarify the epoch length? The figure says -1 to 3 seconds around stimulus onset but the in-text explanation suggests that a 5 second epoch was obtained i.e., -1 to 4 seconds with stimulus onset.
Because of the large reporting deficit pertaining to EEG measures, I would highly recommend the research group review the most current set of guidelines for:
- EEG research, published by Babiloni et al. (2020) in Clinical Neurophysiology, available at: https://www.sciencedirect.com/science/article/pii/S1388245719311642#b0450
- EEG reporting, published by Kane et al. (2017) in Clinical Neurophysiology Practice at: https://www.sciencedirect.com/science/article/pii/S2467981X17300215
I’d also recommend that any future studies adhere to the recommendations outlined by these working groups to ensure study design and methodological quality meets this standard.
It’s still unclear how ‘lost in thought’ was defined operationally. Did you regard no behavioural response as being lost in thought (i.e., a missed response), or was it an incorrect response to the image (i.e., where a Yes response was required, they responded with no)? How did you handle response errors? You’ve reported the total number of trials the participants performed, but what were the remaining number of trials that were submitted for analysis, after artifact removal?
Were the epochs averaged according to the state type before being analysed in Matlab, or were the single trials analysed?
Why did you conduct AUC analyses? And why did these include the frequency ranges exceeding the bands of interest in this study i.e., above 30 Hz?
Why are the behavioural outcomes not reported in the results? I would assume the authors recorded reaction times, and whether responses were correct or incorrect. This would provide useful behavioural insight into the processing efficiency and accuracy of participants, particularly when comparing states.
Figure 3 demonstrates that data were submitted for FFT and averaged but there’s no explanation of this in the method section. Also, how did you group the data from all participants?
Results
This section is structured well and the figures clearly demonstrate the findings that are presented in-text. I’m wondering how the authors “observed that the 30 Hz SSVEP flashing frequency was more comfortable for subjects than the 12 Hz SSVEP frequency”.
The statistical results, however, have not been reported. Where are the exact F and p values for each finding?
It looks like there’s a significant amount of variability in classification accuracy; how did you determine that this was successful?
Discussion
This section does well to integrate the findings of the present study with the existing literature. However, it is noted that much of the research described here is not reviewed in the introduction. This is relevant literature that should be reported in the introduction to provide context for the current study’s design and hypotheses. The discussion section should then compare the current set of findings against these previous studies.
Finally, can the authors really conclude that their classification system was effective, based on a study of 13 participants where the classification average was below 80%? A sample size as small as this would really offer preliminary findings that require confirmation in a larger cohort before being applied to BCI systems.
Author Response

(The authors gave the same response as above.)

Round 2
Reviewer 1 Report
The authors have significantly improved the paper, specialy in clarity and details, following most of my recommendations. However, some of the analysis- or methodology-related suggestions were not taken into account. The fact that other works do something similar does not necessarily implies that it is the most appropriate for the current work. In this sense, the paper is acceptable but the contribution and soundness keep low due to the important limitations, which are now explicit in the text at least.